# Redirecting Research on Fe^0^ for Environmental Remediation: The Search for Synergy

**DOI:** 10.3390/ijerph16224465

**Published:** 2019-11-13

**Authors:** Rui Hu, Chicgoua Noubactep

**Affiliations:** 1School of Earth Science and Engineering, Hohai University, Fo Cheng Xi Road 8, Nanjing 211100, China; rhu@hhu.edu.cn; 2Department of Applied Geology, Universität Göttingen, Goldschmidtstraße 3, D-37077 Göttingen, Germany

**Keywords:** environmental remediation, iron corrosion products, reaction mechanism, water treatment, zero-valent iron

## Abstract

A survey of the literature on using metallic iron (Fe^0^) for environmental remediation suggests that the time is ripe to center research on the basic relationship between iron corrosion and contaminant removal. This communication identifies the main problem, which is based on the consideration that contaminant reductive transformation is the cathodic reaction of iron oxidative dissolution. Properly considering the inherent complexities of the Fe^0^/H_2_O system will favor an appropriate research design that will enable more efficient and sustainable remediation systems. Successful applications of Fe^0^/H_2_O systems require the collective consideration of progress achieved in understanding these systems. More efforts should be made to decipher the long-term kinetics of iron corrosion, so as to provide better approaches to accurately predict the performance of the next generation Fe^0^-based water treatment systems.

## 1. Introduction

The broad research community working on “metallic iron (Fe^0^) for environmental remediation” largely considers Fe^0^ to be an environmental reducing agent. This idea has been repeated in the introduction and discussion sections of hundreds of research and review articles [1,2,3]. The fundamental flaws identified have been hindering the progress of knowledge on designing efficient and sustainable Fe^0^-based water treatment systems [2,4].

The idea that Fe^0^ reduces selected contaminants under operational conditions was introduced in the peer-reviewed literature in 1994 [5] and was a controversial discussion topic for the following five years [6,7,8,9,10,11]. For example, Warren et al. [7] used several elemental metals (Al^0^, Fe^0^, Sn^0^, and Zn^0^) to treat carbon tetrachloride (CT) and reported “unresolved mechanistic complexities”. CT was one of the organic compounds used in the seminal 1994 work [5]. On the other hand, Weber [9] used 4-aminoazobenzene as a probe molecule to “confirm” that “reductive transformation by Fe^0^ is a surface-mediated process”. According to Weber [9], electron transfer at the Fe^0^/H_2_O interface can be facilitated by appropriate “electron mediators”. Similar controversies have been reported regarding the removal mechanism of several inorganic contaminants [6,10,11]. However, in 1998, a “broad consensus” on reductive transformations as main contaminant removal mechanisms in Fe^0^/H_2_O systems was reached [12]. Despite having been proven wrong several times, the reductive transformation concept is still prevalent [13,14,15,16,17,18,19].

Reports disproving the reductive transformation concept are numerous and scattered in the scientific literature. For example, Noubactep et al. [14] demonstrated that there is no uranium removal without “free” precipitating iron corrosion products (FePCs). They used pyrite (a pH-shifting agent) and MnO_2_ (a Fe^2+^-consuming agent) to delay the “free” precipitation of FeCPs in the vicinity of Fe^0^. Ghauch et al. [16] and Gheju and Balcu [19] similarly used MnO_2_ to delay the removal of diclofenac and chromium, respectively. On the other hand, Jiao et al. [15] clearly demonstrated that observed CT-reductive transformation in Fe^0^/H_2_O systems cannot be mediated by the Fe^0^ surface. Finally, Ebelle et al. [18] systematically demonstrated the non-suitability of the experimental design of Matheson and Tratnyek [5] to achieve the often questioned but still largely accepted conclusions.

The named arguments [14,15,16,17] and the demonstration by Ebelle et al. [18] suggest that the popular view that Fe^0^ is an environmental reducing agent should be urgently revisited. Such a profound correction will offer new research perspectives towards the realization of the full potential of Fe^0^-based treatment systems [20]. The perspectives include: (i) a science-based assessment of the limits of Fe^0^-based remediation systems (advantages, disadvantages, and sustainability), (ii) the design of reliable, efficient, and sustainable Fe^0^-based remediation systems, and (iii) the design of programs to educate designers and operators of such systems. Proper capacity building of the treatment processes, the expected performance, and the required operational conditions would enable well-designed, efficient, and sustainable systems. This paper aimed to pave the way to a bright future of Fe^0^-based remediation technology by presenting some key aspects that have been either overseen or not particularly well considered in the past. The discussion is limited to the chemistry of aqueous iron corrosion.

## 2. Method

The approach used herein considers Fe^0^ as a generator of FeCPs, which are (i) contaminant scavengers, (ii) stand-alone reducing agents, and (iii) fouling agents in filtration systems. Readers that are not familiar with this decade-old-concept are referred to the corresponding literature [21,22,23,24,25,26,27]. This approach implies that the Fe^0^ reactivity yielding contaminant-reductive transformation is not an intrinsic property of Fe^0^ (electrons from Fe^0^). Contaminant reduction is associated with primary (Fe^II^, H/H_2_) and secondary FeCPs (e.g., Fe_3_O_4_, green rust) as iron is corroded by water (the solvent) [4,28,29,30]. The existing literature was evaluated as to the contribution it has made to disregarding the content of the concept and the concept itself (from 2007 on). The next sections discuss how the confusion has been kept alive within the Fe^0^ research community in the past three decades.

### 2.1. A Questionable Literature Review

Fe^0^ remediation technology is more than 170 years old [31]. Ideally, the further development of this technology should be limited to assembling performance data on operating Fe^0^/H_2_O systems in order to improve the design of new systems. However, a critical consideration of the history of the Fe^0^ remediation technology reveals that no systematic approach has ever been followed in designing systems [30,32,33]. In the past three decades, an exceedingly pragmatic approach has been followed [3,34,35].

The three steps toward establishment of any technology are (i) a science-based feasibility analysis, (ii) laboratory scale treatability studies of derived designs, and (iii) pilot plant testing. This systematic approach is considered to have virtually been followed while establishing the Fe^0^ technology [36,37,38]. However, the very first step cannot be considered to have occurred for at least three reasons: (i) the literature review was very limited, as several historical aspects were overlooked [2,31,33]; (ii) the nature of Fe^0^ as a long-term source of iron oxides and hydroxides (contaminant scavengers) [39,40] was not really realized, and instead its “electron-producing” property was favored [3,5,9]; and (iii) the volumetric, expansive nature of iron corrosion was overlooked [41,42]. This initial approximate problem analysis is the major problem of the Fe^0^ technology and must be urgently fixed, especially since Fe^0^/H_2_O systems have the potential to secure universal safe drinking water provision in the developing world within one decade [19,20]. Properly considering that Fe^0^ is not a reducing agent under environmental conditions would clarify all discrepancies reported in the literature [4,33], and thus, aid to shape research for the design of better systems.

### 2.2. The Chemistry of the Fe^0^/H_2_O System

Once immersed in water, a reactive Fe^0^ specimen starts its oxidative dissolution to form Fe^II^ species and H_2_ (Equation (1)). Both primary iron corrosion products (Fe^II^ and H_2_) are stand-alone reducing agents. In the presence of oxygen and/or other oxidizing agents (e.g., MnO_2_), Fe^II^ is further oxidized to Fe^III^ (Equation (2)) [14,16,19]. At pH values > 4.5, Fe^II^ and Fe^III^ hydroxides have low solubility; they polymerize and precipitate to various types of oxides and hydroxides (Equation (3)).
Fe^0^ + 2 H^+^ ⇒ Fe^2+^ + H_2_(1)
4 Fe^2+^ + O_2_ + 2 H^+^ ⇒ 4 Fe^3+^ + 2 OH^−^(2)
Fe^3+^ + Fe^2+^ + n H_2_O 4 Fe^2+^ + O_2_ + 2 H^+^ ⇒ 4 Fe^3+^ + 2 OH^−^ Fe^II^, Fe^III^, and mixed oxyhydroxides(3)
Fe^0^ + RX + H^+^ ⇒ Fe^2+^ + RH + X^−^(4) where X represents a halogen ion (e.g., Cl).

In other words, a Fe^0^/H_2_O system is a dynamic system containing potential reducing agents (Fe^0^, Fe^II^, and H_2_) and adsorbents (oxyhydroxides). It is thus not surprising that adsorption and chemical reduction occur in such systems [25,28,29]. It is rather surprising that the chemical reduction observed is considered to be the cathodic process coupled with the Fe^0^ oxidative dissolution (Equation (1)) and described by reactions equivalent to Equation (4) [4,32,33,43]. 

Equation (4) implies that Fe^0^ is oxidized by RX in an electrochemical process. This corresponds to the view that Fe^0^ is a reducing agent, and constitutes the heart of discrepancy [4]. The major prerequisite for this mechanism is that the oxide scale on Fe^0^ is electronically conductive [43,44]. Such a conductive layer has not been reported under environmental conditions. Accordingly, quantitative reduction by electrons from the metal is not likely to occur [4,32,33,43]. The well-documented reduction is thus mediated by other available reducing agents. Consequently, “reduction in the presence of Fe^0^” should not be considered to be “reduction by Fe^0^” [4]. An electrochemical mechanism for O_2_ was already out ruled in the 1980s, as summarized by Stratmann and Müller [45]. Accordingly, (i) Fe^0^ is oxidized by water (Equation (1)) (electrochemical reaction), and (ii) contaminants and O_2_ are reduced by Fe^II^, H_2_, and/or other species like green rust (chemical reaction).

During the precipitation of oxyhydroxides (Equation (3)), dissolved species are mechanically entrapped in their matrix (co-precipitation). The volume of each corrosion product (oxide) is larger than that of the parent metallic iron (V_oxide_ > V_iron_) [41,42,46]. This evidence implies that in a filter bed containing Fe^0^, iron corrosion pursued for contaminant removal is the first cause of porosity reduction and thus permeability loss [4,30,33,35,42]. Properly considering the point outlined here will soon unify the research community on the evidence that a new orientation of research on Fe^0^ is long overdue. It should be explicitly stated that the notion of electron efficiency, defined as the ratio of electrons transferred to the target contaminants, is not acceptable, as it considers electrochemical reduction to be a relevant reduction path [4,47]. The presentation herein has insisted that reducing electrons are from H_2_ (two electrons per mole of Fe^0^) and/or from Fe^II^ species (one electron per mole of Fe^0^). Accordingly, a good starting position is three electrons per mole of Fe^0^ (for pure abiotic reduction). H_2_ escaping from the system is a fraction of the non-utilized amount because H_2_ evolution is not quantitative at pH > 4.5 [48].

### 2.3. The Root Origin of Confusion

A profound analysis of the Fe^0^/H_2_O system (Section 2.2) revealed that the root origin of confusion within the Fe^0^ remediation research community is nothing but the understanding of the corrosion reaction which can be generally given by Equation (4) [49,50]:metal + oxidizing agent ⇒ oxidized metal + reducing agent(5)

For iron, Equation (5) reads: Fe^0^ + oxidizing agent ⇒ Fe^II^/Fe^III^ species + reducing agent(5a)

The questions arises, which relevant oxidizing agents are available for Fe^0^ corrosion under natural conditions? The prevailing approach regards Fe^0^ as a reducing agent with the electrode potential −0.44 V. Accordingly, Fe^0^ should reduce all dissolved species with a higher electrode potential, including O_2_ and contaminants, (E^0^ > −0.44 V). The research community has been trying to address this issue on a case-by-case basis for three decades already, with very limited success [1,3,19,20,36,38,50]. However, this concept does not consider the evidence that in aqueous solutions at pH > 4.5, the Fe^0^ surface is constantly covered with an electrically non-conductive oxide scale which hinders any quantitative reaction after Equation (4) [30,50]. A clear experimental proof against this concept has been the repeated observation of a time lag between the start of the experiment and the start of the reducing reaction [33,35]. If the electrochemical reduction (electrons from Fe^0^) was a relevant reaction mechanism, it would have been quantitative when the Fe^0^ surface is still free from FeCPs [30,50]. The lag time indicates that even acting reducing agents are generated in situ.

In contrast, the approach presented herein considers water (H_2_O, the solvent) to be the sole relevant oxidizing agent for Fe^0^ (Equation (1)). The reason is that the natural oxide scale acts as a diffusion barrier to all dissolved species. It follows that any reasoning using Equation (4) (E^0^ > −0.44 V) has not properly considered the physicochemistry of the Fe^0^/H_2_O system. Thus, trying to consider the stoichiometry of equations similar to Equation (4) in models is simply false. The most obvious damage of such an approach has been the discussion on which species fill the initial porosity of Fe^0^-based filtration systems: oxide scale (FeCPs), H_2_ accumulation, or foreign precipitates (e.g., CaCO_3_, FeCO_3_). This controversy was related best by Henderson and Demond [37,51,52]. These two authors wrote one of the best review articles on “remediation with Fe^0^” [37], confirming permeability loss as one of the major problems of Fe^0^ filters. Their efforts to solve the named problem went through the stoichiometry of reactions similar to Equation (4) [51] and resulted in their recommending FeS-based systems as less-clogging alternatives [52]. However, this approach cannot properly account for H_2_ evolution, as each mole of oxidized Fe^0^ generates one mole of H_2_ occupying a volume more than 31,000 times larger than the volume of the parent Fe^0^. For illustration, Equation (1) has been re-written to show the changes in the number of moles of Fe^0^ and FeCPs between the start of the operation (t_0_ = 0) and Fe^0^ complete depletion (t_∞_):
(6)$$\begin{array}{cccccccc} & {\mathrm{Fe}}^{0} & + & 2\;H_{2}O & \Rightarrow & {\mathrm{Fe}(\mathrm{OH})}_{2} & + & H_{2}\\ t_{0}=0 & n_{0} & & x & & 0 & & 0\\ t>t_{0} & n_{0}\;(1-\alpha ) & & x^{\prime } & & \alpha \;n_{0} & & \alpha \;n_{0}\\ t_{\infty } & 0 & & x^{\prime \prime } & & n_{0} & & n_{0}\\ \mathrm{Volume} & 1 & & \text{-} & & 2.1\;\mathrm{to}\;6.4 & & >3100 \end{array}$$

The values of x, x’, and x” are not relevant for the discussion, as water is the solvent.

The factor 31,000 arises from the evidence that one mole of Fe^0^ (ρ_iron_ = 7.84 g cm^3^) occupying a volume of 7.14 cm^3^ produces one mole of H_2_ occupying a volume of 22,400 cm^3^ (assuming ideal gas behavior). Accordingly, properly reasoning with the changes in volume accompanying aqueous iron corrosion implies that H_2_ must escape or be used for other processes for the system to be sustainable. The mean feature of the reasoning based on Equation (6) is that modeling porosity loss should have started by considering pore filling by expansive FeCPs [42]. Pore filling by the process of iron corrosion has only recently been demonstrated by X-ray computed tomography of systems containing Fe^0^ and deionized water [48,53]. Again, with a proper system analysis, this evidence would have been considered in the early phase of the technology development. Instead, it is still under discussion whether mixing Fe^0^ with non-expansive aggregates (e.g., gravel, pumice, sand) is suitable for efficient systems [48,53].

### 2.4. Is Multidisciplinarity the Problem?

The subtlety and scientific diversity of the physical and chemical processes involved in water treatment using Fe^0^-based systems is demonstrated herein. The application of Fe^0^ materials in water treatment and environmental remediation (“putting corrosion to use” [54]) is, in essence, an engineering task. However, the present work has demonstrated that, despite three decades of intensive research, scientists have not yet established the science of the Fe^0^/H_2_O system. This common effort should be derived from the available basics of corrosion, electrochemistry, and surface chemistry of metals [30,43,49]. While the value of multidisciplinarity had been widely recognized in establishing the science of aqueous iron corrosion [43,44,49], the behavior of active scientists on remediation with Fe^0^ seems to ultimately preclude results.

The current literature on remediation with Fe^0^ can be regarded as a collection of independent research and review articles, with the term “zero-valent iron” in common. It is difficult for a critical reader to identify signs of common effort even from some critical review articles. As an example, while the intrinsic reactivity of Fe^0^ materials has been recognized as a crucial design parameter in the early phase of technology development [55,56], only a few contaminant-independent tools have been presented and none of them have really been adopted [57,58,59,60]. This means that researchers have not even tried to characterize granular materials (mm in size) before manufacturing granular bimetallics and nano-scale materials [30,50]. On the contrary, reactivity testing is continued with species as toxic as ^99^Tc^VII^ [61].

The same pragmatic trend can be observed in all other aspects of efforts to enhance the efficiency of Fe^0^-based systems, of which two are discussed here in some detail: (i) admixing Fe^0^ with pyrite (FeS_2_) or other FeS minerals, and (ii) adding complexing agents to the system. Adding FeS_2_ to a Fe^0^/H_2_O system corresponds to a pH shift to lower values. In such a system, the pH value is controlled by two antagonistic processes: FeS_2_ dissolution, lowering the pH, and Fe^0^ corrosion, enhancing the pH. Because FeS_2_ dissolution is kinetically more favorable, the pH of the system first decreases and then subsequently progressively increases to a final value [14]. If FeS_2_ dissolution initially shifted the pH to values lower that 4.5, quantitative contaminant removal will be observed only after the system’s pH increases to values above 4.5 [14,62]. This observation has been documented the best in quiescent batch experiments. However, most available works have varied the initial pH values and the FeS_2_ mass loading and reported controversial observations, as summarized by Du et al. [63]. Moreover, authors have generally not monitored the final pH values and the experimental vessels were vigorously shaken or stirred to enable suspension of Fe^0^ and FeS_2_, while FeS_2_ aims to support electron transfer from Fe^0^ thanks to its semi-conductive properties.

As concerns complexing agents, they replace water molecules in the hydration sphere of a dissolved metal ion (Fe^2+^ and/or Fe^3+^) and thus increase Fe solubility [49]. Since the seminal work of Matheson and Tratnyek [5], various complexing agents have been commonly used in investigating the mechanism of contaminant removal. Tested ligands include ethylenediaminetetraacetic acid (EDTA) and 1,10-Phenanthroline (Phen). These ligands are used to avoid or delay the passivation of the Fe^0^ surface. However, it has been well-documented that individual ligands modify the redox properties of the two involved redox systems (Fe^II^/Fe^0^ and Fe^III^/Fe^II^) differently [64]. For example, Phen stabilizes Fe^II^ while EDTA stabilizes Fe^III^. It follows that the addition of EDTA favors reduction by Fe^II^, a reaction that could never quantitatively occur in the absence of EDTA [49,64,65,66]. In other words, in trying to keep the Fe^0^ surface free, bias is introduced. Again, investigated systems have been shaken or stirred, but the extent to which the complexing agents are consumed have not considered in discussions of the results.

The enumerated mistakes suggest that multidisciplinary study has been misinterpreted in the Fe^0^ remediation literature for the last 30 years. Exploiting aqueous iron corrosion for water treatment was known before the natures of many pollutants were established [2,31,39]. Thus, successfully using Fe^0^ to remove contaminants from water does not transform any scientist into an “iron corrosion expert”. Corrosion science is a stand-alone branch of science and at least some extra effort is required to become familiar with the basics of the highly complex process of iron corrosion [49]. Having neglected this key evidence, researchers have been hopelessly searching for common underlying mechanisms for reactions in Fe^0^/H_2_O systems “that provide a confidence for design that is non-site-specific” [67]. However, the treatability experiments have been ill-designed (shaken or stirred) and interpreted with incorrect descriptors such as the k_SA_ concept [22,23,24,25].

The authors of the present communication suggest that the past three decades have presented a significant body of data confirming the suitability of Fe^0^-based systems for the remediation for a large number of pollutants. If relevant data on the iron corrosion rate were made available, efficient remediation systems could be designed using a concept presented in 1991 by Boris M. Khudenko [43] and long-term column experiments for filtration systems [68,69]. Khudenko’s concept [43] has been largely ignored in the scientific literature while contra-intuitive views have been massively published. Khudenko’s concept was recently discovered and proven in line with a concept Noubactep and colleagues have presented more than a decade ago [21,22]. Noubactep’s concept has been similarly ignored by active researchers [50,70,71]. The fact that many researchers work with a falsified view should be of concern to the whole scientific community. This is a problem which cannot be resolved by a few individuals [28,29,71]. This communication was written to make the problem better known to the research community.

## 3. Conclusions

Using Fe^0^/H_2_O systems is a suitable alternative to other methods of water treatment and environmental remediation. Testing their suitability has been unfortunately hampered by several flaws, including (i) incomplete literature reviews; (ii) inappropriate system analysis, in particular overlooking the expansive nature of aqueous iron corrosion; (iii) inappropriate testing conditions (shaking and stirring the vessels); and, of course, (iv) incorrect interpretation of good experimental observations. Because contaminant removal and contaminant reduction have generally been randomly interchanged, even species like aromatic amines, which have been claimed to require microbiological treatment after treatment in Fe^0^/H_2_O systems, can be quantitatively removed upon proper design, as reduction is not a relevant removal mechanism. The literature has presented many examples of non-reducible species (including methylene blue) that have been quantitatively removed in Fe^0^/H_2_O systems [72,73,74].

One constant myth has been an argument based on the idea of the Fe^0^ surface as a site accessible or not to contaminants. At the pH range of relevance, quantitative contaminant diffusive transport to the Fe^0^ surface is impossible, especially since there is no driving force. H_2_ and Fe^2+^ extraction from the Fe^0^ surface (kinetic energy) acts against the transport of any species to the Fe^0^ surface. For a better understanding of the operating mode of Fe^0^/H_2_O systems, long-term experiments under near-nature conditions should be performed to characterize the time-dependent kinetics of iron corrosion and their impact of the efficiency of the system. Hence, both the induction period (lag time prior to contaminant reduction) and the decrease in the corrosion rate will be better understood and used to design more efficient and sustainable Fe^0^-based systems. All that is needed is a truly collaborative research effort, respecting the basic scientific principles of research and publication.

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
