# Peer review of "Redirecting Research on Fe0 for Environmental Remediation: The Search for Synergy"

_ijerph, 2019, doi:10.3390/ijerph16224465_

Round 1

Reviewer 1 Report

I think it can be published as a Perspective

Author Response

Many thanks for this evaluation!

Kind regards,

Dr. Noubactep

Reviewer 2 Report

Review comments

I appreciate the updated revision of this manuscript. Most of my comments on the early version have been responded reasonably. My remaining concern is some arbitrary expression. More objective words may be more suitable in such a scientific journal.

Abstract: For the manuscript type has changed from Communication to Perspective, the information present in the Abstract should be more concrete.

L50-51: Which arguments? In which conditions, such a viewpoint should be “abandoned”? From the context of classic inorganic chemistry, “supplement”, or “amendment”, is more reasonable than “abandonment”.

L68: Which literature? How to collect and select such references? More detailed information should be added in the Methods.

L72: The caption can’t cover the contents of this section.

L225-226: This expression is arbitrary and not always the truth. Please add the reference.

L261: “Full of examples”?

Author Response

Reviewer 2

Comments 1: I appreciate the updated revision of this manuscript. Most of my comments on the early version have been responded reasonably. My remaining concern is some arbitrary expression. More objective words may be more suitable in such a scientific journal.

Responses 1: Many thanks for this evaluation, we really appreciate the collaborative elan! Our only goal is the message to the colleagues.

Comments 2: Abstract: For the manuscript type has changed from Communication to Perspective, the information present in the Abstract should be more concrete.

Responses 2: Many the abstract has been revised, the last sentence was deleted. (Only the awareness of the mistaken view would stimulate the adoption of new directions for future research.)

Comments 3: L50-51: Which arguments? In which conditions, such a viewpoint should be “abandoned”? From the context of classic inorganic chemistry, “supplement”, or “amendment”, is more reasonable than “abandonment”.

Responses 3: Many thanks, “abandoned” is replaced by “urgently revisited”; it reads now:

The named arguments and the demonstration by Ebelle et al. [18] suggest that the popular view that Fe0 is an environmental reducing agent should be urgently revisited.

Comments 4: L68: Which literature? How to collect and select such references? More detailed information should be added in the Methods.

Responses 4: The literature considering Fe0 as “generator of FeCPs” as summarized in ref. [21-27] which are all overview articles. No action.

Comments 5: L72: The caption (“A questionable literature review”) can’t cover the contents of this section.

Responses 5: We strongly disagree with the reviewer as this can be regarded main problem of the Fe0 research community. While pioneers (including Gillham 2008) of the technology have reported that there were no works on reduction of RCl by Fe0 in the literature, and have sought longer to find some Japanese contributions that have partly never been translated in English, we are demonstrating that more accessible articles (in tune with the chemistry of the system) were published in parallel (Khudenko 1991). On the other side, we have derived Fe0 for save drinking water provision from reports on Fe0 permeable reactive barriers, while historically, “Fe0 for save drinking water” was the first improvement of slow sand filters.

Comments 6: L225-226: This expression is arbitrary and not always the truth. Please add the reference.

Responses 6: “It follows that the addition of EDTA can favor reduction by FeII, a reaction that could never quantitatively occur in the absence of EDTA [49,64-66].”

[64] Rizvi, M.A.; Syed, R.M.; Khan, B. Complexation effect on redox potential of iron(III)−iron(II) couple: A 419 simple potentiometric experiment. J. Chem. Educ. 2011, 88, 220–222.

[65] Naka, D.; Kim, D.; Strathmann, T.J. Abiotic reduction of nitroaromatic compounds by aqueous iron(II)-catechol complexes. Environ. Sci. Technol. 2006, 40, 3006–3012.

[66] Rizvi, M.A. Complexation modulated redox behavior of transition metal systems. Russian J. General Chem. 2015, 85, 959–973.

We strongly disagree with the reviewer. Each ligand (L) replaces H2O in [Fe(H2O)6]2+ and the chemistry of the resulting system depends on the relative stability of FeL2+ and FeL3+. The electrode potential can be written as function of the stability constant. Prof. Strathmann (Naka and Strathmann) has been working on this in the environmental context for years. This is a textbook knowledge (Landolt 2007) and has been published and reviewed by Rizvi and colleagues. These for references are added to support the assertion.

The reviewer is right that “not always the truth(but it is neverarbitrary”) but when it is not truth, it is just the exception confirming the rule. The classical example of Cu+ and Cu2+ can be cited.

Comments 7: L261: “Full of examples”?

Responses 7: Many thanks we have condidered this important remark and the text reads now:

The literature presents many examples of non reducible species (including methylene blue) that are quantitatively removed in Fe0/H2O systems [70-72].

[70] Lackovic, J.A.; Nikolaidis, N.P.; Dobbs, G.M. Inorganic arsenic removal by zero-valent iron. Environ. Eng. Sci. 2000, 17, 29–39.

[71] Jia, Y.; Aagaard, P.; Breedveld, G.D. Sorption of triazoles to soil and iron minerals. Chemosphere 2007, 67, 250–258.

[72] Frost, R.L.; Xi, Y.; He, H. Synthesis, characterization of palygorskite supported zero-valent iron and its application for methylene blue adsorption. J. Colloid Interface Sci. 2010, 341, 153–161.

Reviewer 3 Report

Dear Authors,

Thank you for your answer.  The manuscript deserves to be published

Author Response

Many thanks for this evaluation!

This manuscript is a resubmission of an earlier submission. The following is a list of the peer review reports and author responses from that submission.

Round 1

Reviewer 1 Report

I think this article is more suitable for “Perspective”.

Authors should present more data to be “Communication” or add more literature study to become a "mini review".

Reviewer 2 Report

Although some points need more evidences or references, I appreciate such ideas and encourage similar works. I think this communication is of importance to related researchers. More scientifically sound expression are recommended instead of the current alarmist protestation

Abstract: Which survey? conducted by the authors or cited from other? The last sentence is an unnecessary trite remark.

L25: this argument is too arbitrary, please justify it.

L35-36: such general remarks are unnecessary, considering the page limit of a communication.

L45: "not a reducing"? missed "agent", or "reductant"?

L51-52: The solubility is not only pH dependent. Actually, the classic Eh-pH stability diagram for iron oxides and hydroxides showed that both Eh and pH control the components in solutions.

L87-90: These sentences are remarks other than conclusions.

Reviewer 3 Report

The paper contains interesting aspects, however, in my opinion, the mistakes in publications related to the role of Fe0 were not that frequent. The opinion in Conclusions saying that: “The fact that most researchers works with a falsified view” seems to be exaggerated. Only a few papers were cited, so it seems that this is not a serious problem in the scientific research community. I did not find a novelty element in this work, and therefore I cannot recommend this work for printing in present form. It seems to me that this topic is not of the main interests to the readers. 

In the referred by the authors of the reviewed manuscript articles, it was important whether iron (Fe0) is an effective factor in reducing nutrients (reduce its concentration in water). The method of using iron has some limitations / disadvantages, which rely on the possibility of creating a layer preventing the reactivity of iron due to its intrinsic passive layer.

The conclusions state: "The fact that most researchers works with a falsified view must be a concern to the whole scientific community". In my opinion it is not clear what false views are referred to. This should be clearly stated. In view of the small number of cited references, it cannot be claimed that the prevailing opinion is widespread. It should be also mentioned that there are many different dissolved chemical compounds in surface water that can possibly react with zero valent iron and Fe ions. I would recommend to develop this thread in this communication.

Summing up, the revised short communication is not very clear. The impression of a prevailing common misconception that “the Fe0 is a reducing agent “ is exaggerated. I consider this information worth of explanation but it requires more precise language.